# Estimating aboveground carbon density and its uncertainty in Borneo's structurally complex tropical forests using airborne laser scanning

Tommaso Jucker[1,2], Gregory P. Asner[3], Michele Dalponte[4], Philip G. Brodrick[3], Christopher D. Philipson[5,6], Nicholas R. Vaughn[3], Yit Arn Teh[7], Craig Brelsford[8], David F.R.P. Burslem[7], Nicolas J. Deere[9], Robert M. Ewers[10], Jakub Kvasnica[11], Simon L. Lewis[12,13], Yadvinder Malhi[14], Sol Milne[7], Reuben Nilus[15], Marion Pfeifer[16], Oliver L. Phillips[12], Lan Qie[10,12], Nathan Renneboog[17], Glen Reynolds[18], Terhi Riutta[10,14], Matthew J. Struebig[9], Martin Svátek[11], Edgar C. Turner[19] and David A. Coomes[1]

[1]Forest Ecology and Conservation group, Department of Plant Sciences, University of Cambridge, Cambridge CB2 3EA, UK
[2]CSIRO Land and Water, 147 Underwood Avenue, Floreat, 6014, Western Australia, Australia
[3]Department of Global Ecology, Carnegie Institution for Science, 260 Panama Street, Stanford, CA 94305 USA
[4]Department of Sustainable Agro-ecosystems and Bioresources, Research and Innovation Centre, Fondazione E. Mach, Via E. Mach 1, 38010 San Michele all'Adige, Italy
[5]Department of Environmental Systems Science, ETH Zürich, Universitätstrasse 16, 8092 Zürich, Switzerland
[6]Centre for Environmental Change and Human Resilience, University of Dundee, Dundee DD1 4HN, UK
[7]School of Biological Sciences, University of Aberdeen, Cruickshank Building, St Machar Drive, Aberdeen, AB24 3UU, UK
[8]Department of Biosciences, Viikki Plant Science Center (ViPS), University of Helsinki, 00014, Helsinki, Finland
[9]Durrell Institute of Conservation and Ecology (DICE), School of Anthropology and Conservation, University of Kent, Canterbury CT2 7NR, UK
[10]Imperial College London, Silwood Park Campus, Buckhusrt Road, Ascot SL5 7PY, UK
[11]Faculty of Forestry and Wood Technology, Department of Forest Botany, Dendrology and Geobiocoenology, Mendel University, Brno, Czech Republic
[12]School of Geography, University of Leeds, Leeds LS2 9JT, UK
[13]Department of Geography, University College London, London WC1E 6BT, UK
[14]Environmental Change Institute, School of Geography and the Environment, University of Oxford, Oxford OX1 3QY, UK
[15]Forest Research Centre, Sabah Forestry Department, P.O. Box 1407, 90715 Sandakan, Sabah, Malaysia
[16]School of Biology, Newcastle University, Newcastle NE1 7RU, UK
[17]Permian Global, Savoy Hill House, 7-10 Savoy Hill, London WC2R 0BU, UK
[18]South East Asia Rainforest Research Partnership (SEARRP), Danum Valley Field Centre, PO Box 60282, 91112 Lahad Datu, Sabah, Malaysia
[19]Department of Zoology, University of Cambridge, Downing Street, Cambridge, CB2 3EJ, UK

*Correspondence to*: David A. Coomes (dac18@cam.ac.uk)

**Abstract.** Borneo contains some of the world's most biodiverse and carbon dense tropical forest, but this 750,000-km² island has lost 62% of its old-growth forests within the last 40 years. Efforts to protect and restore the remaining forests of Borneo hinge on recognising the ecosystem services they provide, including their ability to store and sequester carbon. Airborne Laser Scanning (ALS) is a remote sensing technology that allows forest structural properties to be captured in great detail

across vast geographic areas. In recent years ALS has been integrated into state-wide assessment of forest carbon in Neotropical and African regions, but not yet in Asia. For this to happen new regional models need to be developed for estimating carbon stocks from ALS in tropical Asia, as the forests of this region are structurally and compositionally distinct from those found elsewhere in the tropics. By combining ALS imagery with data from 173 permanent forest plots spanning the lowland rain forests of Sabah, on the island of Borneo, we develop a simple-yet-general model for estimating forest carbon stocks using ALS-derived canopy height and canopy cover as input metrics. An advanced feature of this new model is the propagation of uncertainty in both ALS- and ground-based data, allowing uncertainty in hectare-scale estimates of carbon stocks to be quantified robustly. We show that the model effectively captures variation in aboveground carbon stocks across extreme disturbance gradients spanning tall dipterocarp forests and heavily logged regions, and clearly outperforms existing ALS-based models calibrated for the tropics, as well as currently available satellite-derived products. Our model provides a simple, generalised and effective approach for mapping forest carbon stocks in Borneo, and underpins ongoing efforts to safeguard and facilitate the restoration of its unique tropical forests.

## 1      Introduction

Forests are an important part of the global carbon cycle (Pan et al., 2011), storing and sequestering more carbon than any other ecosystem (Gibbs et al., 2007). Estimates of tropical deforestation rates vary, but roughly 61,300 km$^2$ of forest were lost each year between 2000 and 2012, and an additional 30% were degraded by logging or fire (Asner et al., 2009; Hansen et al., 2013). Forest degradation and deforestation causes substantial releases of greenhouse gases to the atmosphere – about 1-2 billion tonnes of carbon per year – which equates to about 10% of global emissions (Baccini et al., 2012). Even if nations de-carbonise their energy supply chains within agreed schedules, a rise of 2°C in mean annual temperature is unavoidable unless 300 million hectares of degraded tropical forests are protected, and land unsuitable for agriculture is reforested (Houghton et al., 2015). Signatories to the Paris agreement, brokered at COP21 in 2015, are now committed to reducing emissions from tropical deforestation and forest degradation (i.e., REDD+; Agrawal et al., 2011), whilst recognising that these forests also harbour rich biodiversity and support livelihoods for around a billion people (Vira et al., 2015).

Accurate monitoring of forest carbon stocks underpins these initiatives to generate carbon credits through REDD+ and similar forest conservation and climate change mitigation programs (Agrawal et al., 2011). Airborne laser scanning (ALS) has shown particular promise in this regard, because it generates high resolution maps of forest structure from which aboveground carbon density (*ACD*) can be estimated (Asner et al., 2010; Lefsky et al., 1999; Nelson et al., 1988; Popescu et al., 2011; Wulder et al., 2012). The principle of ALS is that laser pulses are emitted downwards from an aircraft, and a sensor records the time it takes for individual beams to strike a surface (e.g., leaves, branches or the ground) and bounce back to the emitting source, thereby precisely measuring the distance between the object and the airborne platform. Divergence of the beam means it is wider than leaves and allows penetration into the canopy, resulting is a 3D point cloud that captures the vertical structure of the forest. By far the most common approach to using ALS data for estimating forest

carbon stocks involves developing statistical models relating *ACD* estimates obtained from permanent field plots to summary statistics derived from the ALS point cloud, such as the mean height of returns or their skew (Zolkos et al., 2013). These 'area-based' approaches were first used for mapping structural attributes of complex multi-layered forests in the early 2000s (Drake et al., 2002; Lefsky et al., 2002), and have since been applied to carbon mapping in several tropical regions (Asner et al., 2010, 2014; Jubanski et al., 2013; Laurin et al., 2014; Réjou-Méchain et al., 2015).

This paper develops a statistical model for mapping forest carbon, and its uncertainty, in Southeast Asian forests. We work with ALS and plot data collected in the Malaysian state of Sabah, on the north-eastern end of the island of Borneo (Fig. 1), which is an important testbed for international efforts to protect and restore tropical forests. Borneo lost around 62% of its old-growth forest in just 40 years as a result of heavy logging, and subsequent establishment of oil palm and forestry plantations (Gaveau et al., 2014, 2016). Sabah lost its forests at an even faster rate in this period (Osman et al., 2012), and because these forests are amongst the most carbon dense in the tropics, carbon loss has been considerable (Carlson et al., 2012a, 2012b; Slik et al., 2010). In response to past and ongoing forest losses, the Sabah state government has recently taken a number of concrete steps towards becoming a regional leader in forest conservation and sustainable management. Among these was commissioning a new high-resolution wall-to-wall carbon map for the entire state, which will inform future forest conservation and restoration efforts across the region. Here we develop the ALS-based model that underpins this new carbon map (Asner et al., 2018).

The approach we take builds on the work of Asner and Mascaro (2014), who proposed a general model for estimating *ACD* (in Mg C ha$^{-1}$) in tropical forests using a single ALS metric – the mean top-of-canopy height (*TCH*, in m) – and minimal field data inputs. The method relates *ACD* to *TCH*, stand basal area (*BA*; in m$^2$ ha$^{-1}$) and the community-weighted mean wood density (*WD*; in g cm$^{-3}$) over a prescribed area of forest such as one hectare, as follows:

$$ACD_{General} = 3.836 \times TCH^{0.281} \times BA^{0.972} \times WD^{1.376} \qquad (1)$$

Asner and Mascaro (2014) demonstrated that tropical forests from 14 regions differ greatly in structure. Remarkably, they found that a generalised power-law relationship could be fitted that transcended these contrasting forests types, once regional differences in structure were incorporated as sub-models relating *BA* and *WD* to *TCH*. However, this general model may generate systematic errors in *ACD* estimates if applied to regions outside the calibration range, and Asner and Mascaro (2014) make clear that regional models should be obtained where possible. Since South East Asian rainforests were not among the 14 regions used to calibrate the general model, and are phylogenetically and structurally distinct from Neotropical and Afrotropical forests (Banin et al., 2012), new regional models are needed before Borneo's forest carbon stocks can be surveyed using ALS. Central to the robust estimation of *ACD* using ALS data is identifying a metric which captures variation in basal area among stands. Asner and Mascaro's (2014) power-law model rests on an assumption that basal area is closely related to top-of-canopy height, an assumption supported in some studies, but not in others (Coomes et al., 2017; Duncanson et al., 2015; Spriggs, 2015). The dominance of Asian lowland rainforests by dipterocarp species make them structurally unique (Banin et al., 2012; Feldpausch et al., 2011; Ghazoul, 2016) and gives rise to greater aboveground carbon

densities than anywhere else in the tropics (Avitabile et al., 2016; Sullivan et al., 2017), highlighting the need for new ALS-based carbon estimation models for this region.

Here we develop a regional model for estimating *ACD* from ALS data that underpins ongoing efforts to map Sabah's forest carbon stocks at high resolution to inform conservation and management decisions for one of the world's most threatened biodiversity hotspots (Asner et al., 2018; Nunes et al., 2017). Building on the work of Asner and Mascaro (2014), we combine ALS data with estimates of *ACD* from a total of 173 permanent forests plots spanning the major lowland dipterocarp forest types and disturbance gradients found in Borneo to derive a simple-yet-general equation for predicting carbon stocks from ALS metrics at hectare resolution. As part of this approach we also develop a novel framework for propagating uncertainty in both ALS- and ground-based data, allowing uncertainty in hectare-scale estimates of carbon stocks to be quantified robustly. To assess the accuracy of this new model, we then benchmark it against existing ALS-derived equations of *ACD* developed for the tropics (Asner and Mascaro, 2014), as well as satellite-based carbon maps of the region (Avitabile et al., 2016; Pfeifer et al., 2016).

## 2 Materials and methods

### 2.1 Study region

The study was conducted in Sabah, a Malaysian state in northern Borneo (Fig. 1a). Mean daily temperature is 26.7 °C and annual rainfall is 2600-3000 mm (Walsh and Newbery, 1999). Severe droughts linked to El Niño events occur about once every ten years (Malhi and Wright, 2004; Walsh and Newbery, 1999). Sabah supports a wide range of forests types, including dipterocarp forests in the lowlands that are among the tallest in the tropics (Fig. 1b; Banin et al., 2012).

### 2.2 Permanent forest plot data

We compiled permanent forest plot data from five forested landscapes across Sabah (Fig. 1a): Sepilok Forest Reserve, Kuamut Forest Reserve, Danum Valley Conservation Area, the Stability of Altered Forest Ecosystems (SAFE) experimental forest fragmentation landscape (Ewers et al., 2011), and Maliau Basin Conservation Area. Here we provide a brief description of the permanent plot data collected at each site, which are summarized in Table 1. Additional details are provided in Supplement S1.

### 2.2.1 Sepilok Forest Reserve

The reserve is a protected area encompassing a remnant of coastal lowland old-growth tropical rainforest (Fox, 1973) and is characterized by three strongly contrasting soil types that give rise to forests that are structurally and functionally very different (Dent et al., 2006; DeWalt et al., 2006; Nilus et al., 2011): alluvial dipterocarp forest in the valleys (hereafter alluvial forests), sandstone hill dipterocarp forest on dissected hillsides and crests (hereafter sandstone forests), and heath forest on podzols associated with the dip slopes of cuestas (hereafter heath forests). We used data from nine permanent 4 ha

forest plots situated in the reserve, three in each forest type. These were first established in 2000–01 and were most recently re-censused in 2013–15. All stems with a diameter ($D$, in cm) ≥ 10 cm were recorded and identified to species (or closest taxonomic unit). Tree height ($H$, in m) was measured for a subset of trees ($n$ = 718) using a laser range finder. For the purposes of this analysis, each 4 ha plot was subdivided into 1-ha subplots, giving a total 36 plots of 1-ha in size. The corners of the plots were geolocated using a Geneq SXBlue II Global Positioning System (GPS) unit, which uses satellite-based augmentation to perform differential correction and is capable of a positional accuracy of less than 2 m (95% confidence intervals).

### 2.2.2 Kuamut Forest Reserve

The reserve is a former logging area that is now being developed as a restoration project. Selective logging during the past 30 years has left large tracts of forest in a generally degraded condition, although the extent of this disturbance varies across the landscape. Floristically and topographically the Kuamut reserve is broadly similar to Danum Valley – with which it shares a western border – and predominantly consists of lowland dipterocarp forests. Within the forest reserve, 39 circular plots with a radius of 30 m were established in 2015–16 spanning a range of forest successional stages, including young secondary forests characterised by the presence of species with low wood densities (e.g., *Macaranga* spp.). Coordinates for the plot centres were taken using a Garmin GPSMAP 64S device with an accuracy of ±10 m (95% confidence intervals). Within each plot, all stems with $D ≥ 10$ cm were recorded and identified to species (or closest taxonomic unit), and $H$ was measured using a laser range finder. Because the radius of the plots was measured along the slope of the terrain (as opposed to a horizontally projected distance), we slope-corrected the area of each plot by multiplying by $\cos(\theta)$, where $\theta$ is the average slope of the plot in degrees as calculated from the digital elevation model obtained from the ALS data. The average plot size after applying this correction factor was 0.265 ha (6% less than if no slope correction had been applied).

### 2.2.3 Danum Valley Conservation Area

The site encompasses the largest remaining tract of primary lowland dipterocarp forest in Sabah. Within the protected area, we obtained data from a 50 ha permanent forest plot which was established in 2010 as part of the Centre for Tropical Forest Science (CTFS) ForestGEO network (Anderson-Teixeira et al., 2015). Here we focus on 45 ha of this plot for which all stems with $D ≥ 1$ cm have been mapped and taxonomically identified (mapping of the remaining 5 ha of forest was ongoing as of January 2017). For the purposes of this study, we subdivided the mapped area into 45 1-ha plots, the coordinates of which were recorded using the Geneq SXBlue II GPS. In addition to the 50 ha CTSF plot, we also secured data from 20 circular plots with a 30 m radius that were established across the protected area by the Carnegie Airborne Observatory (CAO) in 2017. These plots were surveyed following the same protocols as those described previously for the plots at Kuamut in Sect. 2.2.2.

### 2.2.4    SAFE landscape and Maliau Basin Conservation Area

Plot data from three sources were acquired from the SAFE landscape and the Maliau Basin Conservation Area: research plots established through the SAFE project, plots used to monitor riparian buffer zones, and plots from the Global Ecosystem Monitoring (GEM) network (http://gem.tropicalforests.ox.ac.uk). As part of the SAFE project, 166 plots of $25 \times 25$ m in size

were established in forested areas (Ewers et al., 2011; Pfeifer et al., 2016). Plots are organized in blocks which span a land-use intensity gradient, ranging from twice-logged forests that are currently in the early stages of secondary succession within the SAFE landscape, to relatively undisturbed old-growth forests at Maliau Basin (Ewers et al., 2011; Struebig et al., 2013). Plots were surveyed in 2010, at which time all stems with $D \geq 10$ cm were recorded and plot coordinates were taken using a Garmin GPSMap60 device (accurate to within $\pm10$ m; 95% confidence intervals). Of these 166 plots, 38 were re-surveyed in

2014, at which time all stems with $D \geq 1$ cm were recorded and tree heights were measured using a laser range finder. Using these same protocols, a further 48 plots were established in 2014 along riparian buffer zones in the SAFE landscape. As with the SAFE project plots, riparian plots are also spatially clustered into blocks. The small size of the SAFE and riparian plots (0.0625 ha) makes them prone to high uncertainty when modelling carbon stocks from ALS (Réjou-Méchain et al., 2014), especially given the relatively low positional accuracy of the GPS coordinates. To minimize this source of error, we chose to

aggregate individual plots into blocks for all subsequent analyses ($n = 27$, with a mean size of 0.5 ha). Lastly, we obtained data from six GEM plots – four within the SAFE landscape and two at Maliau Basin. The GEM plots are 1-ha in size and were established in 2014. All stems with $D \geq 10$ cm were mapped, measured for height using a laser range finder, and taxonomically identified. The corners of the plots were georeferenced using the Geneq SXBlue II GPS.

### 2.3    Estimating aboveground carbon density and its uncertainty

Across the five study sites we compiled a total of 173 plots that together cover a cumulative area of 116.1 ha of forest. For each of these plots we calculated aboveground carbon density (*ACD*, in Mg C ha$^{-1}$) following the approach outlined in the *BIOMASS* package in R (R Core Development Team, 2016; Réjou-Méchain et al., 2017). This provides a workflow to not only quantify *ACD*, but also propagate uncertainty in *ACD* estimates arising from both field measurement errors and uncertainty in allometric models. The first step is to estimate the aboveground biomass (*AGB*, in kg) of individual trees using

Chave et al.'s (2014) pantropical biomass equation: $AGB = 0.067 \times (D^2 \times H \times WD)^{0.976}$. For trees with no height measurement in the field, *H* was estimated using a locally calibrated *H–D* allometric equations, while wood density (*WD*, in g cm$^{-3}$) values were obtained from the global wood density database (Chave et al., 2009; Zanne et al., 2009; see Supplement S1 for additional details on both *H* and *WD* estimation).

In addition to quantifying *AGB*, Réjou-Méchain et al.'s (2017) workflow uses Monte Carlo simulations to propagate

uncertainty in biomass estimates due to (i) measurement errors in *D* (following Chave et al.'s (2004) approach, where 95% of stems are assumed to contain small measurement errors that are in proportion to *D*, while the remaining 5% is assigned a gross measurement error of 4.6 cm), (ii) uncertainty in *H–D* allometries, (iii) uncertainty in *WD* estimates arising from

incomplete taxonomic identification and/or coverage of the global wood density database, and (iv) uncertainty in the *AGB* equation itself. Using this approach, we generated 100 estimates of *AGB* for each recorded tree. *ACD* was then quantified by summing the *AGB* of all trees within a plot, dividing the total by the area of the plot, and applying a carbon content conversion factor of 0.47 (Martin and Thomas, 2011). By repeating this across all simulated values of *AGB*, we obtained 100
estimates of *ACD* for each of the 173 plots that reflect the uncertainty in stand-level carbon stocks (note that a preliminary analysis showed that 100 iterations were sufficient to robustly capture mean and standard deviation values of plot-level *ACD*, while also allowing for efficient computing times). As a last step, we used data from 45 plots in Danum Valley – where all stems with $D \geq 1$ cm were measured – to develop a correction factor that compensates for the carbon stocks of stems with $D$ < 10 cm that were not recorded (Phillips et al., 1998; see Eq. (S2) in Supplement S1).

**2.3.1    Stand basal area and wood density estimation**

In addition to estimating *ACD* for each plot, we also calculated basal area (*BA*, in m$^2$ ha$^{-1}$) and the community-weighted mean *WD*, as well as their uncertainties. *BA* was quantified by summing $\pi \times (D/2)^2$ across all stems within a plot, and then applying a correction factor that accounts for stems with $D$ < 10 cm that were not measured (see Eq. (S3) in Supplement S1). In the case of *BA*, uncertainty arises from measurement errors in $D$, which were propagated through following the approach
of Chave et al. (2004) described in Sect. 2.3. The community-weighted mean *WD* of each plot was quantified as $\sum BA_{ij} \times WD_i$, where $BA_{ij}$ is the relative basal area of species $i$ in plot $j$, and $WD_i$ is the mean wood density of species $i$. Uncertainty in plot-level *WD* reflects incomplete taxonomic information and/or lack of coverage in the global wood density database.

**2.4    Airborne laser scanning data**

ALS data covering the permanent forest plots described in Sect. 2.2 were acquired through two independent surveys, the first
undertaken by NERC's Airborne Research Facility (ARF) in November of 2014 and the second by the Carnegie Airborne Observatory (CAO) in April of 2016. Table 1 specifies which plots where flown with which system. NERC ARF operated a Leica ALS50-II LiDAR sensor flown on a Dornier 228-201 at an elevation of 1400–2400 m.a.s.l. (depending on the study site) and a flight speed of 120–140 knots. The sensor emits pulses at a frequency of 120 kHz, has a field of view of 12° and a footprint of about 40 cm. The average point density was 7.3 points m$^{-2}$. The Leica ALS50-II LiDAR sensor records both
discrete point and full waveform ALS, but for the purposes of this study only the discrete return data, with up to four returns recorded per pulse, were used. Accurate georeferencing of the ALS point cloud was ensured by incorporating data from a Leica base station running in the study area concurrently to the flight. The ALS data were pre-processed by NERC's Data Analysis Node and delivered in LAS format. All further processing was undertaken using LAStools software (http://rapidlasso.com/lastools). The CAO campaign was conducted using the CAO–3 system, a detailed description of
which can be found in Asner et al. (2012). Briefly, CAO–3 is a custom-designed, dual-laser full-waveform system that was operated in discrete return collection mode for this project. The aircraft was flown at 3600 m.a.s.l. at a flight speed of 120– 140 knots. The ALS system was set to a field of view of 34° (after 2° cut-off from each edge) and a combined-channel pulse

frequency of 200 kHz. The ALS pulse footprint at 3600 m.a.s.l. was approximately 1.8 m. With adjacent flight-line overlap, these settings yielded approximately 2.0 points m$^{-2}$. Despite differences in the acquisition parameters of the two surveys which can influence canopy metrics derived from ALS data (Gobakken and Næsset, 2008; Roussel et al., 2017), a comparison of regions of overlap between the flight campaigns showed strong agreement between data obtained from the two sensors (Supplement S2).

### 2.4.1    Airborne laser scanning metrics

ALS point clouds derived from both surveys were classified into ground and non-ground points, and a digital elevation model (DEM) was fitted to the ground returns to produce a raster at 1 m resolution. The DEM was then subtracted from the elevations of all non-ground returns to produce a normalised point cloud, from which a canopy height model (CHM) was constructed by averaging the first returns. Finally, any gaps in the raster of the CHM were filled by averaging neighbouring cells. From the CHMs we calculated two metrics for each of the permanent field plots: top-of-canopy height (*TCH*, in m) and canopy cover at 20 m aboveground (*Cover$_{20}$*). *TCH* is the mean height of the pixels which make up the surface of the CHM. Canopy cover is defined as the proportion of area occupied by crowns at a given height aboveground (i.e., 1 – gap fraction). *Cover$_{20}$* was calculated by creating a plane horizontal to the ground in the CHM at a height of 20 m aboveground, counting the number of pixels for which the CHM lies above the plane, and then dividing this number by the total number of pixels in the plot. A height of 20 m aboveground was chosen as previous work showed this to be the optimal height for estimating plot-level *BA* in an old-growth lowland dipterocarp forest in Sabah (Coomes et al., 2017).

### 2.4.2    Accounting for geopositional uncertainty

Plot coordinates obtained using a GPS are inevitably associated with a certain degree of error, particularly when working under dense forest canopies. However, this source of uncertainty is generally overlooked when attempting to relate field-estimates of *ACD* to ALS metrics. To account for geopositional uncertainty, we introduced normally-distributed random errors in the plot coordinates. These errors were assumed to be proportional to the operational accuracy of the GPS unit used to geolocate a given plot: ±2 m for plots recorded with the Geneq SXBlue II GPS and ±10 m for those geolocated using either the Garmin GPSMap60 or Garmin GPSMAP 64S devices. This process was iterated 100 times, and at each step we calculated *TCH* and *Cover$_{20}$* across all plots. Note that for plots from the SAFE project and those situated along riparian buffer zones, ALS metrics were calculated for each individual 0.0625 ha plot before being aggregated into blocks (as was done for the field data).

### 2.5    Modelling aboveground carbon density and associated uncertainty

We started by using data from the 173 field plots to fit a regional form of Asner and Mascaro's (2014) model, where *ACD* is expressed as the following function of ALS-derived *TCH* and field-based estimates of *BA* and *WD*:

$$ACD = \rho_0 \times TCH^{\rho_1} \times BA^{\rho_2} \times WD^{\rho_3} \tag{2}$$

where $\rho_{0-3}$ represent constants to be estimated from empirical data. In order to apply Eq. (2) to areas where field data are not available, the next step is to develop sub-models to estimate $BA$ and $WD$ from ALS metrics. Of particular importance in this regard is the accurate and unbiased estimation of $BA$, which correlates very strongly with $ACD$ (Pearson's correlation coefficient ($\rho$) = 0.93 across the 173 plots). Asner and Mascaro (2014) found that a single ALS metric – $TCH$ – could be used to reliably estimate both $BA$ and $WD$ across a range of tropical forest regions. However, recent work suggests this may not always be the case (Duncanson et al., 2015; Spriggs, 2015). In particular, Coomes et al. (2017) showed that ALS metrics that capture information about canopy cover at a given height aboveground – such as $Cover_{20}$ – were better suited to estimating $BA$. Here we compared these two approaches to test whether $Cover_{20}$ can prove a useful metric to distinguish between forests with similar $TCH$ but substantially different $BA$.

### 2.5.1 Basal area sub-models

Asner and Mascaro (2014) modelled $BA$ as the following function of $TCH$:

$$BA = \rho_0 \times TCH \tag{3}$$

We compared the goodness of fit of Eq. (3) to a model that additionally incorporates $Cover_{20}$ as a predictor of $BA$. Doing so, however, requires accounting for the fact that $TCH$ and $Cover_{20}$ are correlated. To avoid issues of collinearity (Dormann et al., 2013), we therefore first modelled the relationship between $Cover_{20}$ and $TCH$ using logistic regression, and used the residuals of this model to identify plots that have higher or lower than expected $Cover_{20}$ for a given $TCH$:

$$\ln\left(\frac{Cover_{20}}{1 - Cover_{20}}\right) = \rho_0 + \rho_1 \times \ln(TCH) \tag{4}$$

Predicted values of canopy cover ($\widehat{Cover}_{20}$) can be obtained from Eq. (4) as follows:

$$\widehat{Cover}_{20} = \frac{1}{1 + e^{-\rho_0} \times TCH^{-\rho_1}} \tag{5}$$

From this, we calculated the residual cover ($Cover_{resid}$) for each of the 173 field plots as $Cover_{20} - \widehat{Cover}_{20}$, and then modelled $BA$ as the following non-linear function of $TCH$ and $Cover_{resid}$ :

$$BA = \rho_0 \times TCH^{\rho_1} \times (1 + \rho_2 \times Cover_{resid}) \tag{6}$$

Eq. (6) was chosen after careful comparison with alternative functional forms. This included modelling $BA$ directly as a function of $Cover_{20}$, without including $TCH$ in the regression. We discarded this last option as $BA$ estimates were found to be highly sensitive to small variations in canopy cover when $Cover_{20}$ approaches 1.

### 2.5.2 Wood density sub-models

Following Asner and Mascaro (2014), we modelled $WD$ as a power-law function of $TCH$:

$$WD = \rho_0 \times TCH^{\rho_1} \qquad\qquad\qquad\qquad (7)$$

The expectation is that, because the proportion of densely-wooded species tends to increase during forest succession (Slik et al., 2008), taller forests should – on average – have higher stand-level *WD* values. While this explicitly ignores the well-known fact that *WD* is also influenced by environmental factors that have nothing to do with disturbance (e.g., soils or climate; Quesada et al., 2012), we chose to fit a single function for all sites as from an operational standpoint applying forest type-specific equations would require information on the spatial distribution of these forest types across the landscape (something which may not necessarily be available, particularly for the tropics). For comparison, we also tested whether replacing *TCH* with *Cover_{20}* would improve the fit of the *WD* model.

### 2.5.3    Error propagation and model validation

Just as deriving accurate estimates of *ACD* is critical to producing robust and useful maps of forest carbon stocks, so too is the ability to place a degree of confidence on the mean predicted values obtained from a given model (Réjou-Méchain et al., 2017). In order to fully propagate uncertainty in ALS-derived estimates of *ACD*, as well as robustly assessing model performance, we developed the following approach based on leave-one-out cross validation: (i) of the 173 field plots, one was set aside for validation, while the rest were used to calibrate models; (ii) the calibration dataset was used to fit both the regional *ACD* model [Eq. (2)], as well as of the *BA* and *WD* sub-models [Eq. (3, 6–7)]; (iii) the fitted models were used to generate predictions of *BA*, *WD* and *ACD* for the validation plot previously set aside. In each case, Monte Carlo simulations were used to incorporate model uncertainty in the predicted values. For Eq. (4) and (6), parameter estimates were obtained using the L-BFGS-B nonlinear optimization routine implemented in Python (Morales and Nocedal, 2011). For power-law models fit to log-log transformed data [i.e., Eq. (2) and (7)], we applied the Baskerville (1972) correction factor by multiplying predicted values by $\exp(\sigma^2/2)$, where $\sigma$ is the estimated standard deviation of the residuals (also known as the residual standard error); (iv) model fitting and prediction steps (ii–iii) were repeated 100 times across all estimates of *ACD*, *BA*, *WD*, *TCH* and *Cover_{20}* that had previously been generated for each field plot. This allowed us to fully propagate uncertainty in *ACD* arising from field measurement errors, allometric models and geopositional errors; (v) lastly, steps (i–iv) were repeated for all 173 field plots.

Once predictions of *ACD* had been generated for all 173 plots, we assessed model performance by comparing predicted and observed *ACD* values (*ACD_{pred}* and *ACD_{obs}*, respectively) on the basis of root mean square error (RMSE) – calculated as $\sqrt{\frac{1}{N}\sum_{i=1}^{N}\left(ACD_{obs} - ACD_{pred}\right)^2}$ – and relative systematic error (or bias), which we calculated as $\frac{1}{N}\sum_{i=1}^{N}\left(\frac{ACD_{pred}-ACD_{obs}}{ACD_{obs}}\right) \times 100$ (Chave et al., 2014). Additionally, we tested how plot-level errors (calculated for each individual plot as $\frac{|ACD_{obs}-ACD_{pred}|}{ACD_{obs}} \times 100$) varied as a function of forest carbon stocks and in relation to plot size (Réjou-Méchain et al., 2014).

The modelling and error propagation framework described above was chosen after a thorough comparison with a number of alternative approaches. The objective of this comparison was to identify the approach that would yield the lowest degree of systematic bias in the predicted values of *ACD*, as we consider this to be a critical requirement of any carbon estimation model, particularly if – as is the case here – that model is to underpin the generation of a carbon map designed to inform

management and conservation policies (Asner et al., 2018). Of the two alternative approaches we tested, the first relied on fitting a combination of ordinary and nonlinear least squares regression models to parametrise the equations presented above. As with the modelling routine described above, this approach did not account for potential spatial autocorrelation in the residuals of the models, which could result in a slight underestimation of the true uncertainty in the fitted parameter values. We contrasted this approach with one that used generalised and nonlinear least squares regression that explicitly account for

spatial dependencies in the data. Both these approaches underperformed compared to the routine described above, as they substantially overestimated *ACD* values in low carbon density forests and underestimated *ACD* in carbon-rich ones (see Supplement S3 for details). This tendency to introduce a systematic bias in the *ACD* predictions was particularly evident in the case of the spatially explicit models (see Fig. S4b). In light of this we opted for the approach presented here, even though we acknowledge that it may slightly underestimate uncertainty in modelled *ACD* values due to spatial non-independence in

the data.

## 2.6     Comparison with satellite-derived estimates of aboveground carbon density

We compared the accuracy of *ACD* estimates obtained from ALS with those of two existing carbon maps that cover the study area. The first of these is a carbon map of the SAFE landscape and Maliau Basin derived from RapidEye satellite imagery (Pfeifer et al., 2016). The map has a resolution of $25 \times 25$ m and makes use of textural and intensity information

from four wavebands to model forest biomass (which we converted to carbon by applying a conversion factor of 0.47; Martin and Thomas, 2011). The second is a recently published consensus map of pan-tropical forest carbon stocks at 1 km resolution (Avitabile et al., 2016). It makes use of field data and high-resolution locally-calibrated carbon maps to refine estimates from existing pan-tropical datasets obtained through satellite observations (Baccini et al., 2012; Saatchi et al., 2011).

To assess the accuracy of the two satellite products, we extracted *ACD* values from both carbon maps for all overlapping field plots and then compared field and satellite-derived estimates of *ACD* on the basis of RMSE and bias. For consistency with previous analyses, *ACD* values for SAFE project plots and those in riparian buffer zones were extracted at the individual plot level (i.e., 0.0625 ha scale) before being aggregated into the same blocks used for ALS-model generation. In the case of Avitabile et al. (2016), we acknowledge that because of the large difference in resolution between the map and

the field plots, comparisons between the two need to be made with care. This is particularly true when only a limited number of field plots are located within a given 1 km$^2$ grid cell. To at least partially account for these difference in resolution when assessing agreement between Avitabile et al.'s (2016) map and the field data, we first averaged *ACD* values from field plots that fell within the same 1 km$^2$ grid cell. We then compared satellite- and plot-based estimates of *ACD* for (i) all grid cells

within which field plots were sampled, regardless of their number and size ($n = 135$) as well as for (ii) a subset of grid cells for which at least five plots covering a cumulative area $\geq 1$ ha were sampled in the field ($n = 8$). The expectation is that grid cells for which a greater number of large plots have been surveyed should show closer alignment between satellite- and plot-based estimates of *ACD*.

## 5   3      Results

The regional model of *ACD* – parameterized using field estimates of wood density and basal area and ALS estimates of canopy height – was:

$$ACD_{Regional} = 0.567 \times TCH^{0.554} \times BA^{1.081} \times WD^{0.186} \qquad (8)$$

The model had an RMSE of 19.0 Mg C ha$^{-1}$ and a bias of 0.6% (Fig. 2a; see Supplement S4 for confidence intervals on parameter estimates for all models reported here). The regional *ACD* model fit the data better than Asner and Mascaro's (2014) general model (i.e., Eq. (1) in the Sect. 1), which had an RMSE of 32.0 Mg C ha$^{-1}$ and tended to systematically underestimate *ACD* values (bias = –7.1%; Fig. 2b).

### 3.1      Basal area sub-models

When modelling *BA* in relation to *TCH*, we found the best-fit model to be:

$$BA = 1.112 \times TCH \qquad (9)$$

In comparison, when *BA* was expressed a function of both *TCH* and *Cover$_{resid}$* we obtained the following model:

$$BA = 1.287 \times TCH^{0.987} \times (1 + 1.983 \times Cover_{resid}) \qquad (10)$$

where $Cover_{resid} = Cover_{20} - \frac{1}{1+e^{12.431 \times TCH^{-4.061}}}$ (Fig. 3). Of the two sub-models used to predict *BA*, Eq. (10) proved the better fit to the data (RMSE = 9.3 and 6.6 m$^2$ ha$^{-1}$, respectively; see Supplement S5), reflecting the fact that in our case *BA* was more closely related to canopy cover than *TCH* (Fig. 4).

### 3.2      Wood density sub-model

When modelling *WD* as a function of *TCH*, we found the best fit model to be:

$$WD = 0.385 \times TCH^{0.097} \qquad (11)$$

Across the plot network *WD* showed a general tendency to increase with *TCH* (Fig. 5; RMSE of 0.056 g cm$^{-3}$). However, the relationship was weak and Eq. (11) did not capture variation in *WD* equally well across the different forest types (see Supplement S5). In particular, heath forests at Sepilok – which have very high *WD* despite being much shorter than surrounding lowland dipterocarp forests (0.64 against 0.55 g cm$^{-3}$) – were poorly captured by the *WD* sub-model. We found

no evidence to suggest that replacing *TCH* with canopy cover at 20 m aboveground would improve the accuracy of these estimates (see Supplement S5).

### 3.3 Estimating aboveground carbon density from airborne laser scanning

When field-based estimates of *BA* and *WD* were replaced with ones derived from *TCH* using Eq. (9) and (11), the regional *ACD* model generated unbiased estimates of *ACD* (bias = –1.8%). However, the accuracy of the model decreased substantially (RMSE = 48.1 Mg C ha$^{-1}$; Fig. 2c). In particular, the average plot-level error was 21% and remained relatively constant across the range of *ACD* values observed in the field data (yellow line in Fig. 6a). In contrast, when the combination of *TCH* and *Cover$_{20}$* were used to estimate *BA* through Eq. (10), we obtained more accurate estimates of *ACD* (RMSE = 39.3 Mg C ha$^{-1}$, bias = 5.3%; Fig. 2d). Moreover, in this instance plot-level errors showed a clear tendency to decrease in large and high-carbon density plots (blue line in Fig. 6a), declining from an average 25.0% at 0.1-ha scale to 19.5% at 0.25-ha, 16.2% at 0.5-ha and 13.4% at 1-ha (blue line in Fig. 6b).

### 3.4 Comparison with satellite-derived estimates of aboveground carbon density

When compared to ALS-derived estimates of *ACD*, both satellite-based carbon maps of the study area showed much poorer agreement with field data (Fig. 7). Pfeifer et al.'s (2016) map covering the SAFE landscape and Maliau Basin systematically underestimated *ACD* (bias = –36.9%) and had an RMSE of 77.8 Mg C ha$^{-1}$ (Fig. 7a). By contrast, Avitabile et al.'s (2016) pan-tropical map tended to overestimate carbon stocks. When we compared field and satellite estimates of *ACD* across all grid cells for which data was available we found that carbon stocks were overestimate by 111.2% on average, with an RMSE of 100.1 Mg C ha$^{-1}$ (grey circles in Fig. 7b). As expected, limiting this comparison to grid cells for which at least five plots covering a cumulative area ≥ 1 ha were sampled led to greater agreement between field and satellite estimates of *ACD* (large black circles in Fig. 7b). Yet the accuracy of the satellite-derived estimates of *ACD* remained much lower than that derived from ALS data (RMSE = 82.8 Mg C ha$^{-1}$; bias = 59.3%).

### 4 Discussion

We developed an area-based model for estimating aboveground carbon stock from ALS data that can be applied to mapping the lowland tropical forests of Borneo. We found that adding a canopy cover term to estimate *BA* to Asner and Mascaro's (2014) general model substantially improved its goodness of fit (Fig. 2c–d), as it allowed us to capture variation in stand basal area much more effectively compared to models parameterized solely using plot-averaged *TCH*. In this process, we also implemented an error propagation approach that allows various sources of uncertainty in *ACD* estimates to be incorporated into carbon mapping efforts. In the following sections we place our approach in the context of ongoing efforts to use remotely sensed data to monitor forest carbon stocks, starting with ALS-based approaches and then comparing these

to satellite-based modelling. Finally, we end by discussing the implication of this work for the conservation of Borneo's forests.

## 4.1 Including canopy cover in the Asner and Mascaro (2014) carbon model

We found that incorporating a measure of canopy cover at 20 m aboveground in the Asner and Mascaro (2014) model improves its goodness-of-fit substantially without compromising its generality. Asner and Mascaro's (2014) model is grounded in forest and tree geometry, drawing its basis from allometric equations for estimating tree aboveground biomass such as that of Chave et al. (2014), where a tree's biomass is expressed as a multiplicative function of its diameter, height and wood density: $AGB = \rho_0 \times (WD \times D^2 \times H)^{\rho_1}$. By analogy, the carbon stock within a plot is related to the product of mean wood density, total basal area and top-of-canopy height (each raised to a power). Deriving this power-law function from a knowledge of the tree size distribution and tree-biomass relationship is far from straightforward mathematically (Spriggs, 2015; Vincent et al., 2014), but this analogy seems to hold up well in a practical sense. When fitted to data from 14 forest types spanning aridity gradients in the Neotropics and Madagascar, Asner and Mascaro (2014) found that a single relationship applied to all forests types, once regional differences in structure were incorporated as sub-models relating $BA$ and $WD$ to $TCH$. However, the model's fit depends critically on there being a close relationship between $BA$ and $TCH$, as $BA$ and $ACD$ tend to be tightly coupled ($\rho = 0.93$ in our case). Whilst that held true for the 14 forest types previously studied, in Bornean forest we found that the $BA$ sub-models could be improved considerably by including canopy cover as an explanatory variable – particularly when it came to estimating $BA$ in densely-packed stands. This makes intuitive sense if one considers an open forest comprised of just a few trees – the crown area of each tree scales with its basal area, so the gap fraction at ground level of a plot is negatively related to the basal area of its trees (Singh et al., 2016). A similar principle applies in denser forests, but in forests with multiple tiers formed by overlapping canopies such as those that occur in Borneo, the best-fitting relationship between gap fraction and basal area is no longer at ground level, but is instead further up the canopy (Coomes et al., 2017). Meyer et al. (2018) recently came to a very similar conclusion, showing that the cumulative crown area of emergent trees estimated at a height of 27 m aboveground using ALS data was strongly and linearly related to $ACD$ across a diverse range of Neotropical forest types.

The functional form used to model $BA$ in relation to $TCH$ and residual forest cover (i.e., Eq. (10) presented above) was selected for two reasons: first, for a plot with average canopy cover for a given $TCH$, the model reduces to the classic model of Asner and Mascaro (2014), making comparisons straightforward. Secondly, simpler functional forms (e.g., ones relating $BA$ directly to $Cover_{20}$) were found to have very similar goodness-of-fit, but predicted unrealistically high $ACD$ estimates for a small fraction of pixels when applied to mapping carbon across the landscape. This study is the first to formally introduce canopy cover into the modelling framework of Asner and Mascaro (2014), but several other studies have concluded that gap fraction is an important variable to include in multiple regression models of forest biomass (Colgan et al., 2012; Meyer et al., 2018; Ni-Meister et al., 2010; Pflugmacher et al., 2012; Singh et al., 2016). Regional calibration of the Asner and Mascaro (2014) model was necessary for the lowland forests of Southeast Asia, because dominance by dipterocarp species make them

structurally unique (Ghazoul, 2016; Jucker et al., 2018): trees in the region grow tall but have narrow stems for their height (Banin et al., 2012; Feldpausch et al., 2011), creating forests that have among the greatest carbon densities of any in the tropics (Avitabile et al., 2016; Sullivan et al., 2017).

The structural complexity and heterogeneity of Sabah's forests is one reason why even though accounting for canopy cover substantially improved the accuracy of our model (particularly in the case of tall, densely packed stands), a certain degree of error remains in the *ACD* estimates (Fig. 6). This error also reflects an inevitable trade-off between striving for generality and attempting to maximise accuracy when modelling *ACD* using ALS. In this regard, our modelling framework differs from the multiple-regression-with-model-selection approach that is typically adopted for estimating the *ACD* of tropical forests using ALS data (Chen et al., 2015; Clark et al., 2011; D'Oliveira et al., 2012; Drake et al., 2002; Hansen et al., 2015; Ioki et al., 2014; Jubanski et al., 2013; Réjou-Méchain et al., 2015; Singh et al., 2016). These studies – which build on two decades of research in temperate and boreal forests (Lefsky et al., 1999; Nelson et al., 1988; Popescu et al., 2011; Wulder et al., 2012) – typically calculate between 5 and 25 summary statistics from the height distribution of ALS returns and explore the performance of models constructed using various combinations of those summary statistics as explanatory variables. The "best-supported" model is then selected from the list of competing models on offer, by comparing relative performance using evaluation statistics such as $R^2$, RMSE or AIC.

There is no doubt that selecting regression models in this way provides a solid basis for making model-assisted inferences about regional carbon stocks and their uncertainty (Ene et al., 2012; Gregoire et al., 2016). However, a well-recognised problem is that models tend to be idiosyncratic by virtue of local fine-tuning, so cannot be applied more widely than the region for which they were calibrated, and cannot be compared very easily with other studies. For example, it comes as no surprise that almost all publications identify mean height or some metric of upper-canopy height (e.g., 90[th] or 99[th] percentile of the height distribution) as being the strongest determinant of biomass. But different choices of height metric make these models difficult to compare. Other studies have included variance terms or measures of laser penetration to the lower canopy in an effort to improve goodness of fit. For instance, a combination of 75[th] quantile and variance of return heights proved effective in modelling *ACD* of selectively logged forests in Brazil (D'Oliveira et al., 2012). Similarly, a model developed for lower montane forests in Sabah included the proportion of last returns within 12 m of the ground (Ioki et al., 2014), while the proportions of returns in various height tiers were selected for ALS carbon mapping of sub-montane forest in Tanzania (Hansen et al., 2015). Working with Asner and Mascaro's (2014) power-law model may well sacrifice goodness-of-fit compared with these locally-tuned multiple regression models. However, it provides a systematic framework for ALS modelling of forest carbon stocks which we expect will prove hugely valuable for calibrating and validating the next generation of satellite sensors being designed specifically to monitor the world's forests.

## 4.2    Quantifying and propagating uncertainty

One of the most important applications of *ACD*-estimation models is to infer carbon stocks within regions of interest. Carbon stock estimation has traditionally been achieved by networks of inventory plots, designed to provide unbiased estimates of

timber volumes within an acceptable level of uncertainty, using well-established design-based approaches (Särndal et al., 1992). Forest inventories are increasingly supported by the collection of cost-effective auxiliary variables, such as ALS-estimated forest height and cover, that increase the precision of carbon stock estimation when used to construct regression models, which are in turn used to estimate carbon across areas where the auxiliary variables have been measured (e.g.,

McRoberts et al., 2013). But just as producing maps of our best estimates of carbon stocks across landscapes is critical to informing conservation and management strategies, so too is the ability to provide robust estimates of uncertainty associated with these products (Réjou-Méchain et al., 2017).

Assessing the degree of confidence which we place on a given estimate of *ACD* requires uncertainty to be quantified and propagated through all process involved in the calculation of plot-level carbon stocks and statistical model fitting (Chen et

al., 2015). Our Monte Carlo framework allows field-measurement errors, geo-positional errors and model uncertainty to be propagated in a straightforward and robust manner (Yanai et al., 2010). Our approach uses Réjou-Méchain et al.'s (2017) framework as a starting point for propagating errors associated with field measurements (e.g., stem diameter recording, wood density estimation) and allometric models (e.g., height-diameter relationships, tree biomass estimation) into plot-level estimates of *ACD*. We then combine these sources of uncertainty with those associated with co-location errors between field

and ALS data and propagate these through the regression models we develop to estimate *ACD* from ALS metrics. This approach – which is fundamentally different to estimating uncertainty by comparing model predictions to validation field plots – is not widely used within the remote sensing community (e.g., Gonzalez et al., 2010) despite being the more appropriate technique for error propagation when there is uncertainty in field measurements (Chen et al., 2015).

Nevertheless several sources of potential bias remain. Community-weighted wood density is only weakly related to ALS

metrics and is estimated with large errors (Fig. 5). The fact that wood density cannot be measured remotely is well recognised, and the assumptions used to map wood density from limited field data have major implications for carbon maps produced by satellites (Mitchard et al., 2014). For Borneo, it may prove necessary to develop separate wood density sub-models for estimating carbon in heath forests versus other lowland forest types (see Fig. 5). Height estimation is another source of potential bias (Rutishauser et al., 2013): four published height-diameter curves for Sabah show similar fits for

small trees (< 50 cm diameter) but diverge for large trees (Coomes et al., 2017), which contain most of the biomass (Bastin et al., 2015). Terrestrial laser scanning is likely to address this issue in the coming years, providing not only new and improved allometries for estimating tree height, but also much more robust field reference estimates of *ACD* from which to calibrate ALS-based models of forest carbon stocks (Calders et al., 2015; Gonzalez de Tanago Menaca et al., 2017). As this transition happens, careful consideration will also need to be given to differences in acquisition parameters among ALS

campaigns and how these in turn influence *ACD* estimates derived from ALS metrics. While we found strong agreement between canopy metrics derived from the two airborne campaigns (Supplement S2), previous work has highlighted how decreasing ALS point density and changing footprint size can impact on the retrieval of canopy parameters (Gobakken and Næsset, 2008; Roussel et al., 2017). In this regard new approaches designed to explicitly correct for differences among ALS flight specifications (e.g., Roussel et al., 2017) offer great promise for minimizing this source of bias.

Lastly, another key issue influencing uncertainty in *ACD* estimates derived from ALS data is the size of the field plots used to calibrate and validate prediction models. As a rule of thumb, the smaller the field plots the poorer the fit between field estimates of *ACD* and ALS-derived canopy metrics (Asner and Mascaro, 2014; Ruiz et al., 2014; Watt et al., 2013). Aside from the fact that small plots inevitably capture a greater degree of heterogeneity in *ACD* compared to larger ones (leading to

more noise around regression lines), they are also much more likely to suffer from errors associated with poor alignment between airborne and field data, as well as exhibiting strong edge effects (e.g., large trees whose crowns straddle the plot boundary). As expected, for our best-fitting model of *ACD* we found that plot-level errors tended to decrease with plot size (blue curve in Fig. 6b), going from 25.0% at 0.1-ha scale to 13.4% for 1-ha plots. This result is remarkably consistent with previous theoretical and empirical work conducted across the tropics, which reported mean errors of around 25-30% at 0.1-

ha scale and approximately 10-15% at 1-ha resolution (Asner and Mascaro, 2014; Zolkos et al., 2013). These results have led to the general consensus that 1-ha plots should become the standard for calibrating against ALS data. That being said, because there is a trade-off between the number of plots one can establish and their size, working with 1-ha plots inevitably comes at the cost of replication and representativeness. As such, in some cases it may be preferable to sacrifice some precision (e.g., by working with ¼-ha plots, which in our case had a mean error of 19.5%) in order to gain a better

representation of the wider landscape – so long as uncertainty in *ACD* is fully propagate throughout.

### 4.3     Comparison with satellite-derived maps

Our results show that when compared to independent field data, existing satellite products systematically under- or over-estimate *ACD* (depending on the product; Fig. 7). While directly comparing satellite-derived estimates with independent field data in not entirely straight forward – particularly when the resolution of the map is much coarser than that of the field

plots (Réjou-Méchain et al., 2014), as is the case with Avitabile et al. (2016) – it does appear that ALS is able to provide much more robust and accurate estimates of *ACD* and its heterogeneity within the landscape than what is possible with current satellite sensors. This is unsurprising given that in contrast to optical imagery which only captures the surface of the canopy, ALS data provides high-resolution information on the 3D structure of canopies which directly relates to *ACD*. However, ALS data is limited in its temporal and spatial coverage, due to high operational costs. Consequently, there is a

growing need to focus on fusing ALS-derived maps of *ACD* with satellite data to advance our ability to map forest carbon stocks across large spatial scales and through time (e.g., Asner et al., 2018). In this regard, NASA plans to start making high resolution laser ranging observations from the international space station in 2018 as part of the GEDI mission, while ESA's BIOMASS mission will use P-band synthetic aperture radar to monitor forests from space from 2021. Pan-tropical monitoring of forest carbon using data from a combination of space-borne sensors is fast approaching, and regional carbon

equations derived from ALS data such as the one we develop here will be critical to calibrate and validate these efforts.

# 5        Conclusions

Since the 1970s Borneo has lost more than 60% of its old-growth forests, the majority of which have been replaced by large-scale industrial palm oil plantations (Gaveau et al., 2014, 2016). Nowhere else has this drastic transformation of the landscape been more evident than in the Malaysian state of Sabah, where forest clearing rates have been among the highest across the entire region (Osman et al., 2012). Certification bodies such as the Roundtable on Sustainable Palm Oil (RSPO) have responded to criticisms by adopting policies that prohibit planting on land designated as High Conservation Value (HCV), and have recently proposed to supplement the HCV approach with High Carbon Stock (HCS) assessments that would restrict expansion of palm oil plantations onto carbon-dense forests. Yet enforcing these policies requires an accurate and spatially-detailed understating of how carbon stocks are distributed cross the entire state, something which is currently lacking. With the view of halting further deforestation of carbon-dense old-growth forests and generating the necessary knowledge to better manage its forests into the future, in 2016 the Sabah state government commissioned CAO to deliver a high-resolution ALS-based carbon map of the entire state (Asner et al., 2018). The regional carbon model we develop here underpins this initiative (Asner et al., 2018; Nunes et al., 2017), and more generally will contribute to ongoing efforts to use remote sensing tools to provide solutions for identifying and managing the more than 500 million ha of tropical lands that are currently degraded (Lamb et al., 2005).

## Data availability

The authors confirm that data supporting the results of this manuscript will be archived and made freely available on the NERC-EICD website, and that the corresponding DOI will be included at the end of the article.

## Author contribution

D.A.C. and Y.A.T. coordinated the NERC airborne campaign, while G.P.A. led the CAO airborne surveys of Sabah. T.J. and D.A.C. designed the study, with input from G.P.A. and P.B.; T.J., M.D., P.B. and N.V. processed the airborne imagery, while other authors contributed field data; T.J. analysed the data, with input from D.A.C, G.P.A., P.B. and C.D.P.; T.J. wrote the first draft of the manuscript, with all other authors contributing to revisions.

## Competing interests

The authors declare that they have no conflict of interest.

## Acknowledgments

This study was funded by the UK Natural Environment Research Council's (NERC) Human Modified Tropical Forests research programme (grant numbers: NE/K016377/1 and NE/K016407/1 awarded to the BALI and LOMBOK consortiums, respectively). We are grateful to NERC's Airborne Research Facility and Data Analysis Node for conducting the survey and pre-processing the airborne data, and to Abdullah Ghani for manning the GPS base station. D.A. Coomes was supported in part by an International Academic Fellowship from the Leverhulme Trust. The Carnegie Airborne Observatory portion of the study

was supported by the UN Development Programme, Avatar Alliance Foundation, Roundtable on Sustainable Palm Oil, World Wildlife Fund, and the Rainforest Trust. The Carnegie Airborne Observatory is made possible by grants and donations to G.P. Asner from the Avatar Alliance Foundation, Margaret A. Cargill Foundation, David and Lucile Packard Foundation, Gordon and Betty Moore Foundation, Grantham Foundation for the Protection of the Environment, W. M. Keck Foundation, John D.

5    and Catherine T. MacArthur Foundation, Andrew Mellon Foundation, Mary Anne Nyburg Baker and G. Leonard Baker Jr, and William R. Hearst III. The SAFE project was supported by the Sime Darby Foundation. We acknowledge the SAFE management team, Maliau Basin Management Committee, Danum Valley Management Committee, South East Asia Rainforest Research Partnership, Sabah Foundation, Benta Wawasan, the State Secretary, Sabah Chief Minister's Departments, Sabah Forestry Department, Sabah Biodiversity Centre and the Economic Planning Unit for their support, access to the field sites and

10   for permission to carry out fieldwork in Sabah. Field data collection at Sepilok was supported by an ERC Advanced Grant (291585, T-FORCES) awarded to O.L. Phillips, who is also a Royal Society-Wolfson Research Merit Award holder. M. Svátek was funded through a grant from the Ministry of Education, Youth and Sports of the Czech Republic (grant number: INGO II LG15051) and J. Kvasnica was funded through an IGA grant (grant number: LDF_VP_2015038). We are grateful to the many field assistants who contributed to data collection.

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

**Table 1:** Summary of permanent forest plot data collected at each study site and description of which ALS sensor was used to at each location. Plot size is in hectares, while minimum stem diameter thresholds ($D_{min}$) are given in cm.

| Study site | Census year | № plots | Plot size | № trees | $D_{min}$ | Height | Species ID | ALS sensor |
|---|---|---|---|---|---|---|---|---|
| Sepilok Forest Reserve | 2013–15 | 36 | 1 | 22430 | 10 | ✓ | ✓ | NERC ARF |
| Kuamut Forest Reserve | 2015–16 | 39 | 0.265[*] | 5588 | 10 | ✓ | ✓ | CAO–3 |
| Danum Valley Conservation area | | | | | | | | |
| *CTFS plot* | 2010–16 | 45 | 1 | 215016 | 1 | ✓ | ✓ | NERC ARF |
| *CAO plots* | 2017 | 20 | 0.271[*] | 2771 | 10 | ✓ | ✓ | CAO–3 |
| SAFE landscape | | | | | | | | |
| *SAFE experiment* | 2014 | 38[†] | 0.0625 | 8444 | 1 | ✓ | | NERC ARF |
| *SAFE experiment* | 2010 | 101[†] | 0.0625 | 2517 | 10 | | | NERC ARF |
| *Riparian buffers* | 2014 | 48[†] | 0.0625 | 1472 | 10 | ✓ | | NERC ARF |
| *GEM plots* | 2014 | 4 | 1 | 1900 | 10 | ✓ | ✓ | NERC ARF |
| Maliau Basin Conservation area | | | | | | | | |
| *SAFE experiment* | 2010 | 27[†] | 0.0625 | 894 | 10 | | | NERC ARF |
| *GEM plots* | 2014 | 2 | 1 | 905 | 10 | ✓ | ✓ | NERC ARF |

[*]Mean plot size after applying slope correction (see Sect. 2.2.2 for further details)

[†]Plots established as part of the SAFE experiment and those located along riparian buffer zones in the SAFE landscape were

5 aggregated into spatial blocks prior to statistical analyses ($n = 27$ with a mean plot size of 0.5 ha; see Sect. 2.2.4 for further details).

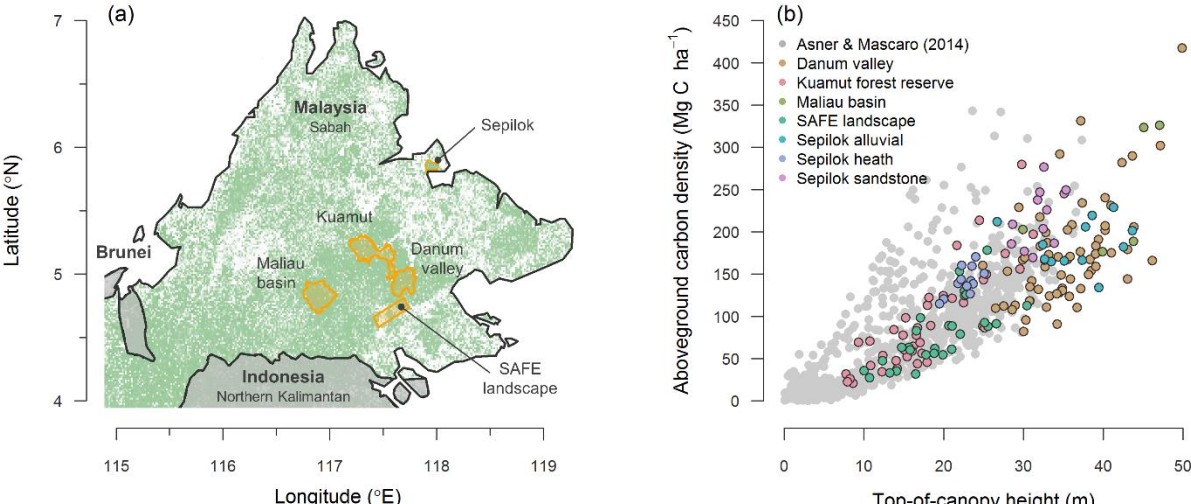

**Figure 1:** Panel (a) shows the location of the Sepilok and Kuamut Forest Reserves, the Danum Valley and Maliau Basin Conservation Areas, and the SAFE landscape within Sabah (Malaysia). Green shading in the background represents forest cover at 30-m resolution in the year 2000 (Hansen et al., 2013). In panel (b), the relationship between field-measured aboveground carbon density and ALS-derived top-of-canopy height found across the study sites (coloured circles, $n = 173$) is compared to measurements taken mostly in the Neotropics (Asner and Mascaro 2014; grey circles, $n = 754$).

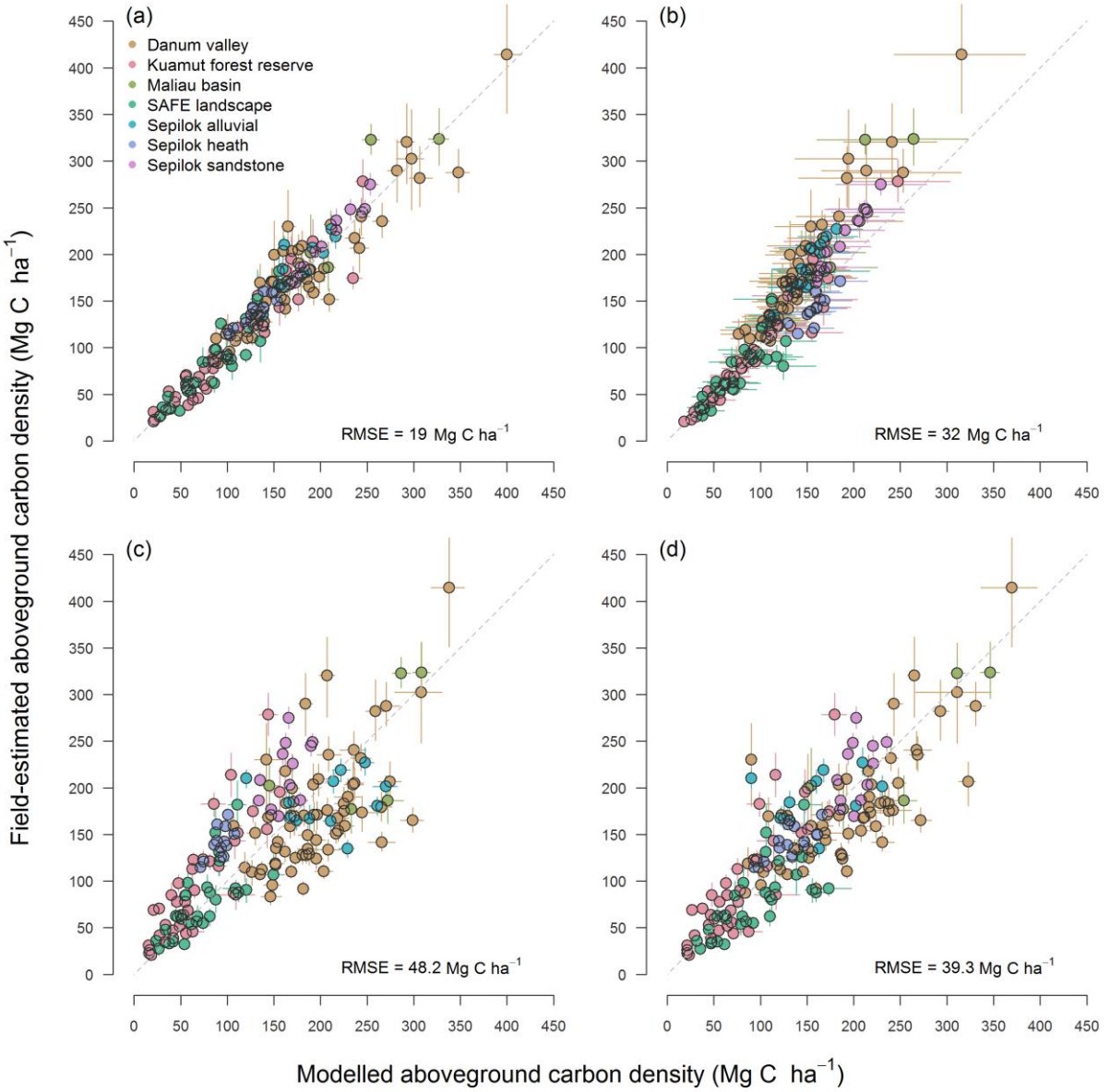

**Figure 2:** Relationship between field-estimated and modelled aboveground carbon density (*ACD*). Panel (a) shows the fit of the regionally-calibrated *ACD* model (Eq. (8) in Sect. 3) which incorporate field-estimated basal area (*BA*) and wood density (*WD*), while (b) corresponds to Asner and Mascaro's (2014) general *ACD* model (Eq. (1) in Sect. 1). Panels (c–d) illustrate the predictive accuracy of the regionally-calibrated *ACD* model when field-measured *BA* and *WD* values are replaced with estimated derived from airborne laser scanning. In (c) *BA* and *WD* were estimated from top-of-canopy height (*TCH*) using Eq. (9) and (11), respectively. In contrast, *ACD* estimates in panel (d) were obtained by modelling *BA* as a function of both *TCH* and canopy cover at 20 meters aboveground following Eq. (10). In all panels, predicted *ACD* values are based on leave-one-out cross validation. Dashed lines correspond to a 1:1 relationship. Error bars are standard deviations and the RMSE of each comparison is printed in the bottom right-hand corner of the panels.

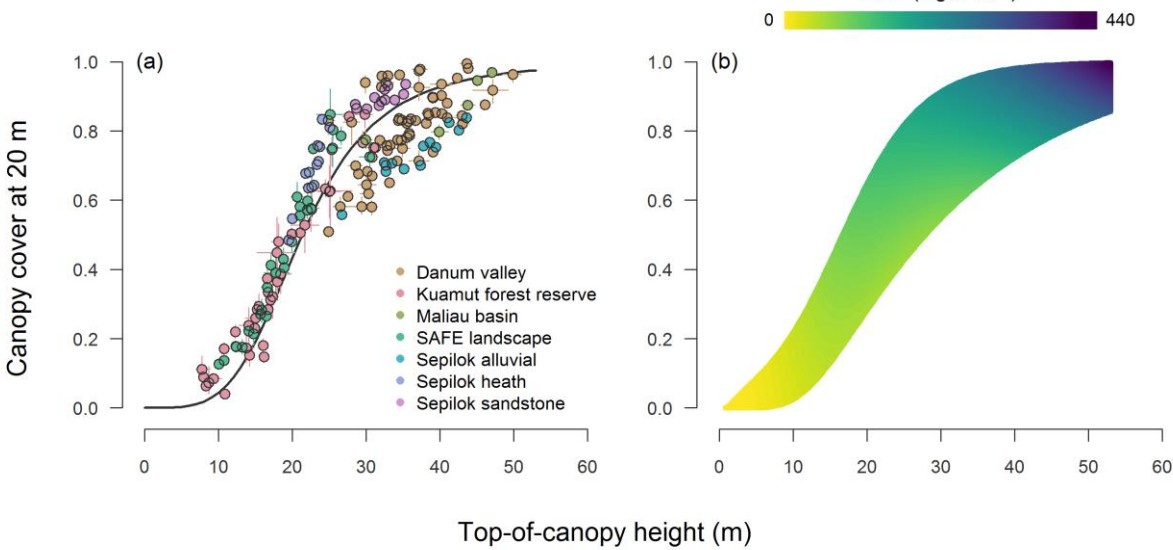

**Figure 3:** Relationship between ALS-derived canopy cover at 20 meters aboveground and top-of-canopy height. Panel (a) shows the distribution of the field plots with a line of best fit passing through the data, with error bars corresponding to standard deviations. Panel (b) illustrates how estimates of aboveground carbon density (*ACD*; obtained using Eq. (8), with Eq. (10) and (11) as inputs) vary as a function of the two ALS metrics for the range of values observed across the forests of Sabah.

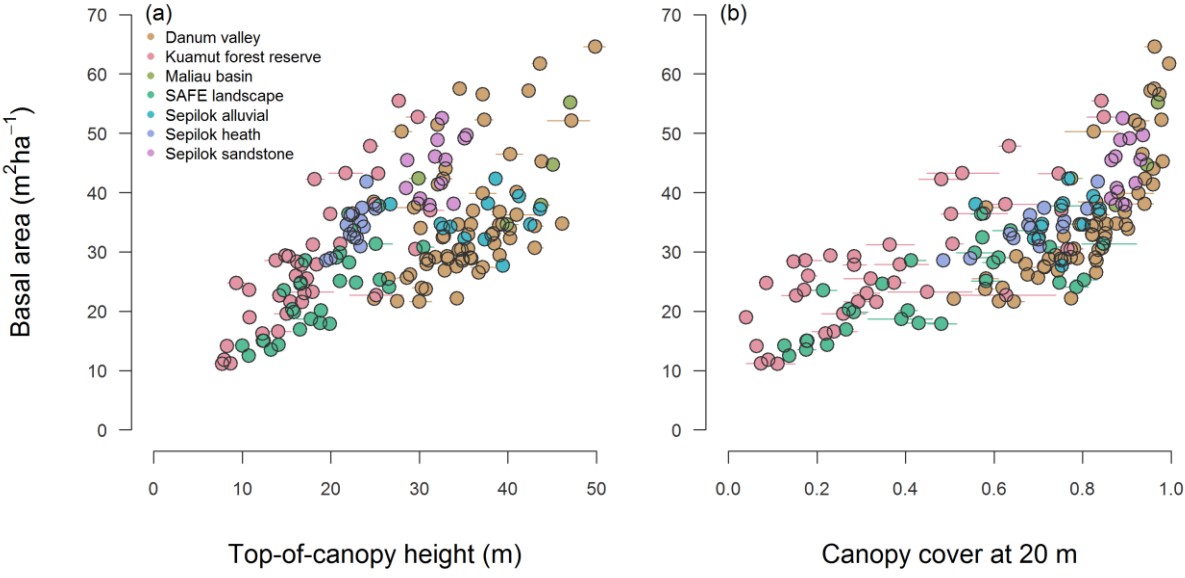

**Figure 4:** Relationship between field-measured basal area and (a) top-of-canopy height and (b) canopy cover at 20 meters aboveground as measured through airborne laser scanning. Error bars correspond to standard deviations.

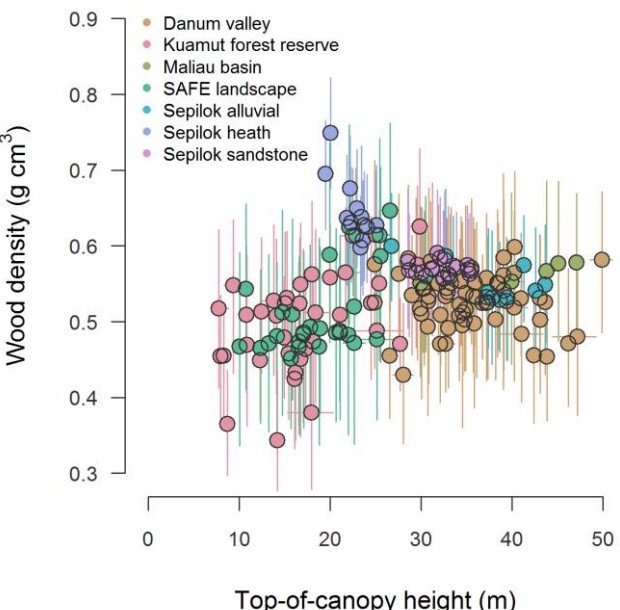

**Figure 5:** Relationship between community-weighted mean wood density (from field measurements) and top-of-canopy height (from airborne laser scanning). Error bars correspond to standard deviations.

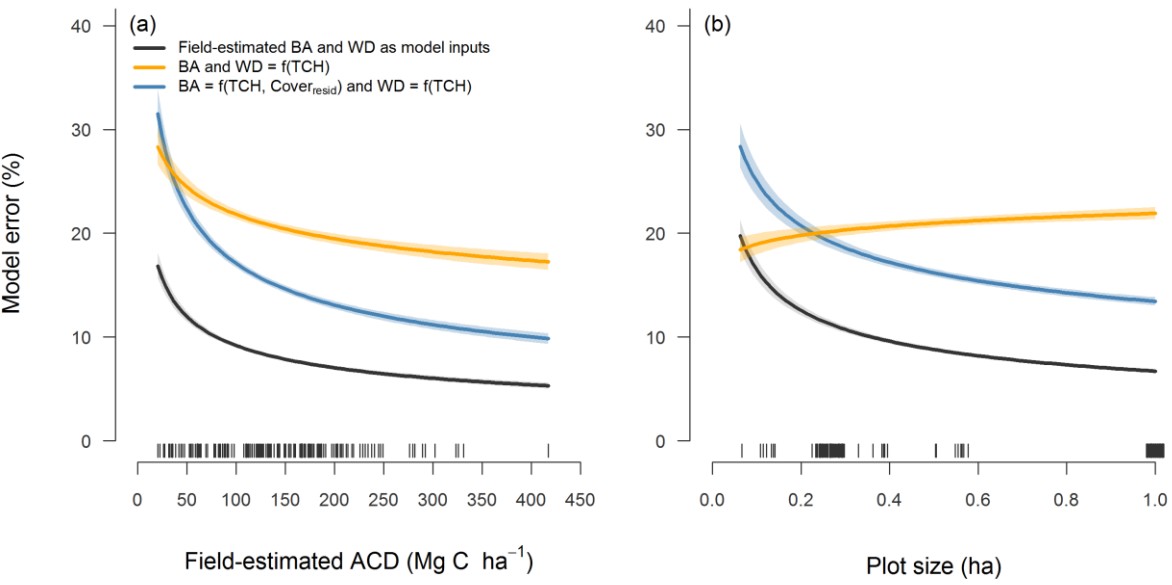

5    **Figure 6:** Model errors (calculated for each individual plot as $\left(\left|ACD_{obs} - ACD_{pred}\right|\right)/ACD_{obs} \times 100$) in relation to (a) field-estimated aboveground carbon density (*ACD*) and (b) plot size. Curves ($\pm$ 95% shaded confidence intervals) were obtained by fitting linear models to log-log transformed data. Black lines correspond to the regionally-calibrated *ACD* model (Eq. (8) in Sect. 3). Orange lines show model errors when basal area (*BA*) was estimated from top-of-canopy height (*TCH*) using Eq. (9). In contrast, blue lines show model errors when *BA* was expressed as a function of both *TCH* and canopy cover at 20 meters aboveground following Eq. (10). Vertical dashed lines along
10   the horizontal axis show the distribution of the data (in panel (b) plot size values were jittered to avoid overlapping lines).

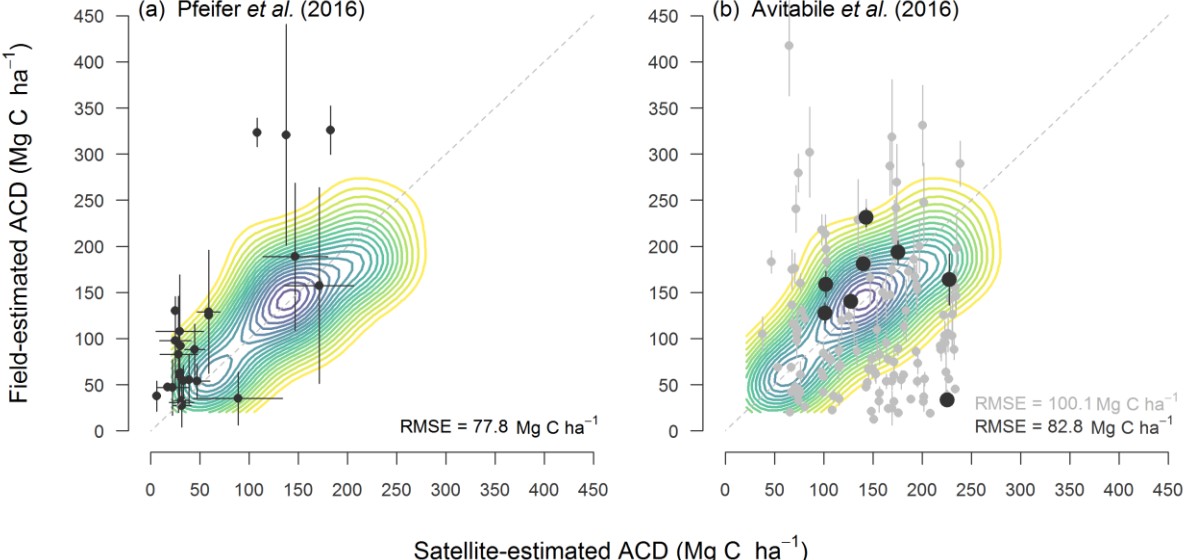

**Figure 7:** Comparison between field-estimated aboveground carbon density (*ACD*) and satellite-derived estimates of *ACD* reported in (a) Pfeifer et al. (2016) and (b) Avitabile et al. (2016). In panel (b) large black point correspond to grid cells in Avitabile et al.'s (2016) pan-tropical biomass map for which at least five plots covering a cumulative area ≥ 1 ha were sampled in the field. By contrast, grid cells for which comparisons are based on less than five plots are depicted by small grey circles. Error bars correspond to standard deviations, while the RMSE of the satellite estimates is printed in the bottom right-hand corner of the panels (note that for panel (b) the RMSE in grey is that calculated across all plots, whereas that in black is based only on the subset of grid cells for which at least five plots covering a cumulative area ≥ 1 ha were sampled in the field). For comparison with *ACD* estimates obtained from airborne laser scanning, a kernel density plot fit to the points in Fig. 2d is displayed in the background.