# Peer review of "Estimating aboveground carbon density and its uncertainty in Borneo's structurally complex tropical forests using airborne laser scanning"

_Biogeosciences, 2018_

## Referee Comment (RC1) · Anonymous Referee #1 · 8 Apr 2018

Dear authors,

Thank you very much for your interesting and meaningful paper regarding large area forest biomass mapping using ALS based on an ecological general model. I feel that the paper is worth to be published. However some explanations should be added to make the meaning or value of results clearer to the readers. Please read my comments below.

Yours sincerely,

General comment: I agree to authors idea about necessity of a general model for ACD mapping. Since the regression approach for ACD prediction is basically a case study in

specific forests, proposing a general model is very important to avoid effort of modeling in each case. Tropical forests have complex species, structure with wide validation and it is difficult to estimate biomass accurate. Therefore this is an important paper which showed a semi-empirical general ACD estimation model using ALS.

I also remind that although the proposed models are based on Asner and Mascaro (2014), the adaptation using ALS parameters is based on regression analysis. Therefore the authors should refer necessity of developing any thorough general models in the future. For example Fig. 2c, d and Fig. 5a suggest that there is a clear relationship between modeled and field ACD among plots less than ca. 70 Mg C ha-1, however, the relationship diverse greatly among plots exceeding it. Fig 5a shows TCH has a clear near-linear relationship with canopy cover among plots less than ca. 25m of TCH, however, it diverse greatly after ca. 25m. These suggest that unknown factors which are caused by canopy changes during tree growth influence on the ACD estimation in Fig 2d. To reduce the prediction errors in Fig. 2d, you have some approaches which are maybe statistical (regression) analysis, semi-empirical or physical modeling. You should suggest the approach to improve accuracy of your general models in the future to readers.

Individual comment: I feel that the paper is documented very well. I suggest adding a few explanations below to improve your paper.

1. P13 25 I suppose that you had better change the following part. adding a canopy cover term to Asner and Mascaro's (2014) general model → adding a canopy cover term to estimate BA in Asner and Mascaro's (2014) general model

2. p27 Fig 4 Three plots in Kuamut forest reserve appeared at the bottom of figure. The authors had better describe about the cause of spread from other plots. The reviewer suppose that they are young secondary forests with pioneer species with small WD. If you have any information about species composition of the three plots and describe them, it will be helpful information to the readers.

3. Fig 2 c, d vs. Fig 7 The authors compare their ACD estimation using ALS and two ACD estimation using satellite optical imagery. The nature or principle of the two systems is different and it cause better performance of ALS based approach than optical image analysis. You should describe about it and make clear the reason of advantage using ALS data.

4. Many reference paper is not shown in the reference list. Please finish the reference list.

END

---

## Referee Comment (RC2) · Anonymous Referee #2 · 13 Apr 2018

Dear editor and authors, This paper deals with the estimation of aboveground biomass in the tropical forest of Borneo Island using the model and airborne laser scanning. I think that this paper is very innovative and important for the evaluation of the carbon stock of the tropical forest. The model of this study can estimate aboveground carbon density (ACD) well, but I'd like to request one to authors. Coomes et al.(2017) reported that ACD is closely related to basal area than to tree height. However, canopy cover at 20m and top-of-canopy height by airborne laser scanning were not good correlation with basal area (Fig.3). I understand that data from airborne laser scanning is not enough and we have to know the basal area and woody density to estimate ACD by using your model. Please give us some suggestion to estimate the ACD by only airborne

laser scanning data in the future. If authors are possible, please add explanation for the difference of representativeness of the data between field observation and airborne laser scanning. This manuscript is nicely ordered, but order of the figure number is not correct, especially page 11 (order is Fig.2, Fig.5 (line 385), Fig.3. . .), please correct it.
* * *

---

## Author Comment (AC1) · 8 May 2018

R1 comment: Dear authors, Thank you very much for your interesting and meaningful paper regarding large area forest biomass mapping using ALS based on an ecological general model. I feel that the paper is worth to be published. However some explanations should be added to make the meaning or value of results clearer to the readers. Please read my comments below. Yours sincerely.

Response: Thank you for taking the time to review our manuscript; we are very pleased you found it of interest. We have made an effort to address your suggestions in the revised version of our manuscript and feel that doing so has helped improve the clarity

of the paper.

R1 comment: General comment: I agree to authors idea about necessity of a general model for ACD mapping. Since the regression approach for ACD prediction is basically a case study in specific forests, proposing a general model is very important to avoid effort of modelling in each case. Tropical forests have complex species, structure with wide validation and it is difficult to estimate biomass accurate. Therefore this is an important paper which showed a semi-empirical general ACD estimation model using ALS.

Response: Thank you for the positive feedback.

R1 comment: I also remind that although the proposed models are based on Asner and Mascaro (2014), the adaptation using ALS parameters is based on regression analysis. Therefore the authors should refer necessity of developing any thorough general models in the future. For example Fig. 2c, d and Fig. 5a suggest that there is a clear relationship between modeled and field ACD among plots less than ca. 70 Mg C ha-1, however, the relationship diverse greatly among plots exceeding it. Fig 5a shows TCH has a clear near-linear relationship with canopy cover among plots less than ca. 25m of TCH, however, it diverse greatly after ca. 25m. These suggest that unknown factors which are caused by canopy changes during tree growth influence on the ACD estimation in Fig 2d. To reduce the prediction errors in Fig. 2d, you have some approaches which are maybe statistical (regression) analysis, semi-empirical or physical modeling. You should suggest the approach to improve accuracy of your general models in the future to readers.

Response: We agree that it is important to acknowledge that – in absolute terms – the magnitude of the model prediction errors tends to be greatest in forests with high aboveground carbon densities (ACD). That being said, it is also important to keep in mind that this pattern is to be expected to some extent – if nothing else because prediction errors are very unlikely to be large, in absolute terms, when plot-level ACD values

are low. This is why we chose to compare the errors of the different models on a proportional scale (see Fig. 6 in the main text). What this shows is that relying exclusively on top-of-canopy height (TCH) for estimating ACD results in a model with errors that are approximately 20% of the mean irrespective of a plot's ACD (orange curve in Fig. 6a). By contrast, when we incorporated a canopy cover term in the modelling framework, the proportional error of the model showed a clear tendency to decrease for plots with higher ACD (blue curve in Fig. 6a). As for the relationship between canopy cover at 20 m aboveground (Cover_20) and TCH, while this exhibits a near-linear trend for TCH values between about 15-30 m, at both ends of the distribution the relationship becomes clearly non-linear. This again is entirely to be expected given the bounded nature of the Cover_20 metric (which cannot be less than 0 or exceed 1). Once TCH drops below 20 m then Cover_20 will rapidly decrease to 0. Similarly, as TCH exceeds 35-40 m, on average Cover_20 tends to quickly approach 1. In revising our paper we have made an effort to clarify that model errors – in absolute terms – have a tendency to be greatest in forests with high ACD. In terms of reducing these errors, as the referee points out there are number of potential approaches that could be used. Arguably the most common of these is to rely on a multiple-regression-with-model-selection approach. This involves calculating a large number of summary statistics from the height distribution of the LiDAR returns and then exploring the performance of models constructed using various combinations of those summary statistics as explanatory variables. While this approach may well reduce predictions errors compared to models calibrated using Asner and Mascaro's (2014) framework, this comes at the cost of producing models that are highly specific to the site for which they were calibrated for. In the Discussion of our paper we argue that while Asner and Mascaro's (2014) approach inevitably sacrifices some goodness-of-fit compared with locally tuned multiple regression models, it provides a systematic framework for large-scale monitoring of forest carbon using LiDAR. This is particularly relevant given the upcoming launch of NASA's GEDI mission, a spaceborne LiDAR sensor designed specifically for monitoring forest ecosystems.

R1 comment: Individual comment: I feel that the paper is documented very well. I

suggest adding a few explanations below to improve your paper.

Response: Thank you. We have worked to incorporate your suggestions into the revised text.

R1 comment: P13 25 I suppose that you had better change the following part adding a canopy cover term to Asner and Mascaro's (2014) general model → adding a canopy cover term to estimate BA in Asner and Mascaro's (2014) general model.

Response: Agreed. We have changed this to clarify that the canopy cover term was used to estimate basal area.

R1 comment: p27 Fig 4 Three plots in Kuamut forest reserve appeared at the bottom of figure. The authors had better describe about the cause of spread from other plots. The reviewer suppose that they are young secondary forests with pioneer species with small WD. If you have any information about species composition of the three plots and describe them, it will be helpful information to the readers.

Response: Yes, plots within the Kuamut Forest Reserve span a range of forest successional stages, including young secondary forests dominated by species with low wood densities (e.g., species of the genus Macaranga). We have clarified this in Section 2.2.2. of the Methods where we describe the Kuamut plots.

R1 comment: Fig 2 c, d vs. Fig 7 The authors compare their ACD estimation using ALS and two ACD estimation using satellite optical imagery. The nature or principle of the two systems is different and it cause better performance of ALS based approach than optical image analysis. You should describe about it and make clear the reason of advantage using ALS data.

Response: Agreed. In the revised paper we have made an effort to clarify that comparisons between the LiDAR-derived and satellite-based estimates of ACD need to be made with care – accounting not only for differences between LiDAR and optical sensors, but also ones related to the spatial grain at which ACD is estimated. What our

analysis shows, unsurprisingly, is that compared to optical imagery, LiDAR is better suited to capturing canopy structural metrics that are strong predictors of ACD. This is particularly evident when comparing our results to the ACD estimates derived from Pfeifer et al. (2016) map (Fig. 7a), which used RapidEye multispectral imagery to estimate ACD at a comparable spatial scale to our analysis and using some of the same plot data for calibration. Nonetheless, while LiDAR clearly has certain advantages when it comes to fine-scale modelling of ACD in tropical forests, optical satellite data are still critical to large-scale monitoring as they provide a spatio-temporal coverage that airborne LiDAR simply cannot match. In this respect we see the two approaches as strongly complementary of one another. Indeed, the recently published carbon map of Sabah's forests (Asner et al. 2018) – which is underpinned by the modelling framework developed in our paper – replied on Landsat imagery to upscale carbon stock estimates generated from airborne LiDAR data acquired over representative areas of the state. We have clarified this in the revised manuscript.

R1 comment: Many reference paper is not shown in the reference list. Please finish the reference list.

Response: Our apologies for this, it was due to an issue with the reference manager software we used. We have resolved this in the revised version of the manuscript and have checked to make sure all cited papers appear in the bibliography.

---

## Author Comment (AC2) · 8 May 2018

R2 comment: Dear editor and authors, This paper deals with the estimation of aboveground biomass in the tropical forest of Borneo Island using the model and airborne laser scanning. I think that this paper is very innovative and important for the evaluation of the carbon stock of the tropical forest. The model of this study can estimate aboveground carbon density (ACD) well, but I'd like to request one to authors. Coomes et al. (2017) reported that ACD is closely related to basal area than to tree height. However, canopy cover at 20m and top-of-canopy height by airborne laser scanning were not good correlation with basal area (Fig.3). I understand that data from airborne

laser scanning is not enough and we have to know the basal area and woody density to estimate ACD by using your model. Please give us some suggestion to estimate the ACD by only airborne laser scanning data in the future. If authors are possible, please add explanation for the difference of representativeness of the data between field observation and airborne laser scanning.

Response: Thank you for reviewing and helping us improve our manuscript, we are pleased you found it of interest. Regarding the estimation of carbon stocks directly from LiDAR, our results clearly show that while attempts to generate general equations that can be applied across forest types are promising, in order to obtain accurate and unbiased estimates of carbon stocks these equations need to be calibrated locally with field data (see comparison between fig 2a and 2b in the main text). In the revised manuscript we go into more detail regarding some of the pros and cons of using Asner and Mascaro's (2014) approach for estimating aboveground carbon density (ACD) from LiDAR. We recognise that by focusing on one (or in our case two) LiDAR metrics for estimating ACD, Asner and Mascaro's (2014) approach may well sacrifice goodness-of-fit compared with locally tuned multiple regression models that incorporate many more LiDAR metrics. However, by doing so the derived models have the virtue of being more applicable and generalizable to other forest types. Looking forward, what our results suggest is that by developing a library of locally-calibrated versions of Asner and Mascaro's (2014) model that adequately capture underlying variation in forest basal area, we will approach a point where variation among forest types is characterised well enough to allow ACD to be estimated directly from LiDAR with little or no need for calibration with field data.

R2 comment: This manuscript is nicely ordered, but order of the figure number is not correct, especially page 11 (order is Fig.2, Fig.5 (line 385), Fig.3), please correct it.

Response: We have corrected this, thank you.